# Cervical cancer screening knowledge and associated factors among Eswatini women: A cross-sectional study

**Phinda G. Khumalo**[1,2]*, **Mariko Carey**[2,3], **Lisa Mackenzie**[1,2], **Rob Sanson-Fisher**[1,2]

**1** Health Behaviour Research Collaborative, School of Medicine and Public Health, College of Health, Medicine, and Wellbeing, The University of Newcastle, Callaghan, NSW, Australia, **2** Hunter Medical Research Institute, New Lambton, NSW, Australia, **3** Centre for Women's Health Research, School of Medicine and Public Health, College of Health, Medicine, and Wellbeing, The University of Newcastle, Callaghan, NSW, Australia

* Phinda.Khumalo@uon.edu.au

**Data Availability Statement:** In accordance with the ethical and legal requirements outlined in the Eswatini Health and Human Research Ethics Guidelines – 3rd Edition, the data supporting the

## Abstract

### Background

Over recent years, cervical cancer incidence and related mortality have steadily increased in Eswatini. Low cervical cancer screening uptake partly explains the situation. Cervical cancer screening-related knowledge is positively associated with screening uptake. Little is known about women's cervical cancer screening-related knowledge in Eswatini.

### Objective

This study aimed to assess cervical cancer screening knowledge and associated factors among Eswatini women eligible for screening.

### Methods

A cross-sectional study involving three hundred and seventy-seven women aged 25 to 59 selected from four primary healthcare clinics in Eswatini was conducted. A paper and pen survey assessed knowledge about cervical cancer risk factors, benefits of screening, the meaning of screening results, recommended screening intervals, and socio-demographics. Descriptive analyses were performed to assess participants' sociodemographic characteristics. Linear regression was applied to examine associations between cervical cancer screening-related knowledge and participants' sociodemographic characteristics.

### Results

Two hundred and twenty-nine (61%) participants answered 80% or more knowledge questions correctly. Compared to HIV-positive participants, HIV-negative participants had 0.61 times lower cervical cancer screening knowledge scores (β = -0.39, 95% CI: -0.56, -0.19, p = 0.03). Participants who travelled more than 30 minutes to the clinic had 0.3 times lower cervical cancer screening knowledge scores (β = -0.70, 95% CI: -1.15, -0.25, p < 0.01) compared to participants who travelled less than 30 minutes to the clinic.

study findings cannot be shared publicly. Data are available from the Eswatini Health and Human Research Review Board (prm2@ehhrrb.org.sz) for researchers who meet the criteria for access to confidential data.

**Funding:** Open Access funding was supported by a project grant from the National Health and Medical Research Council/Global Alliance for Chronic Diseases ([PGK], [MC], [LM], [RSF]; APP2010271).

**Competing interests:** The authors have declared that no competing interests exist.

## Conclusions

Relatively high overall cervical cancer screening knowledge levels were observed among the study participants. Findings from the current study may inform future educational programs to create and sustain an accurate understanding of cervical cancer screening in Eswatini communities.

## Introduction

Over recent years, the number of Swati women with cervical cancer has steadily increased [1]. In 2018, Eswatini became the country with the world's highest age-standardised incidence rate (84.5 per 100 000 female population) [2]. Approximately 6.5% of women develop cervical cancer before turning 75 years [3]. Mortality trends are similar to incidence trends in the country, with cervical cancer-related mortality estimated at 55.7 per 100 000 female population [2].

The World Health Organisation (WHO) recommends a comprehensive cervical cancer prevention program that involves reducing human papillomavirus (HPV) infections through vaccination and detection and treatment of cervical pre-cancer lesions through screening [4]. Without a vaccination program in Eswatini, screening remains the primary method of cervical cancer prevention. The Ministry of Health in Eswatini has implemented an opportunistic cervical cancer screening program that relies on women presenting at primary health clinics for screening. Visual inspection with acetic acid (VIA) is the screening method of choice. Other techniques, like cytology-based screening and HPV testing, are currently not feasible because of a shortage of financial, infrastructure, human resources and technology investments required to sustain such methods [5]. Unfortunately, the uptake of cervical cancer screening is generally low in Eswatini. A cross-sectional study conducted in 2017 found that only 5.2% of women of ages 30 to 65 years old reported having ever screened [6].

Previous research suggests that knowledge is positively associated with screening uptake [7–9]. Knowledge of cervical cancer risk factors and the benefits of cervical cancer screening is associated with higher screening participation rates [10, 11]. Correct information on these aspects helps develop positive cervical cancer screening attitudes and perceptions that influence whether a woman participates in cancer screening [12]. Also, the probability of promptly seeking screening and follow-up care may be higher among women informed about recommended screening intervals and the meaning of screening results [13, 14].

Little is known about women's cervical cancer screening-related knowledge levels in Eswatini. Ngwenya and Huang's study assessed cervical cancer screening knowledge among Swati men and women. It was found that 58.1% of the participants had misconceptions about the causes of cervical cancer. For example, a majority believed that cervical cancer is a disease caused by witchcraft, cervical cancer is caused by having back street abortion, only women with multiple partners get cervical cancer, and that cervical cancer is a disease for rich people only [6].

The Eswatini cervical cancer screening guidelines recommend that women participate in screening between the ages of 25 and 59. Ngwenya and Huang's study only included women 30 – 59 years old [6]. Therefore, its results may not be representative of women eligible for screening in Eswatini. Also, the scope of knowledge domains in the study was limited to knowledge regarding symptoms and risk factors. Other important domains (such as knowledge of screening eligibility, benefits of cervical cancer screening and the meaning of cervical cancer screening results) previously assessed in other African studies [15, 16] were not

examined. Therefore, the current study seeks to provide improvements on these methodological weaknesses.

In another study, 78% of women aged 18–69 years attending clinics in three regions of Eswatini reported having never heard of VIA [17]. Despite half the sample being HIV-positive, the study did not examine knowledge about the different screening intervals recommended for HIV-positive and HIV-negative women. Eswatini guidelines recommend that HIV-positive women are screened yearly due to their increased risk of cervical cancer, while HIV-negative women are screened every two years.

The current study was conducted to assess the level of knowledge regarding cervical cancer and screening among women aged 25 to 59 years in Eswatini. The study considered knowledge domains, including risk factors of cervical cancer, recommended screening intervals, benefits, and the meaning of cervical cancer screening results. The study also explored the extent to which participants' sociodemographic characteristics are associated with knowledge level. We hypothesised that age, being married, education, HPV-positive status, socioeconomic status (estimated using electricity availability), and the number of clinic visits in the last six months would be positively correlated with knowledge and travel time to the nearest clinic negatively correlated. This study's findings may inform future educational programmes aimed at increasing knowledge levels among women in Eswatini.

## Materials and methods

### Design and setting

A cross-sectional survey was conducted from October to December 2021 among women aged 25 to 59 from four selected primary healthcare clinics in Eswatini. Located in Southern Africa, Eswatini shares its borders with South Africa and Mozambique [18]. Eswatini is a lower middle-income country with a per capita Gross Domestic Product (GDP) of US$2,776 [19] and a population of 1.2 million people [20]. Much of the population lives in rural areas, with only 22% in urban areas [19]. The population of Eswatini is generally young, with 47% aged less than 18 years [21]. About half of the Emaswati are female (52.6%) [21]. Healthcare services are delivered in both private and public sectors. Public healthcare is organised in a four-tiered system. The system consists of a network of community-based services, primary healthcare facilities, secondary healthcare facilities (health centres and regional hospitals), and tertiary healthcare facilities (referral hospitals) [22]. Primary healthcare clinics provide first-line curative and emergency interventions to the rural population. They also offer promotive and preventative services [23]. Among other preventive services, primary healthcare clinics in Eswatini offer health screening, including cervical cancer screening [24]. HPV vaccination is not available in Eswatini [25].

Clinics were eligible if they provided care to at least 50 women aged 25 to 59 per week. Clinics that exclusively served HIV-positive women were not eligible to participate in the study. At the time of data collection, 94 clinics attended to more than 50 women per week [26]. One eligible clinic from each of the four regions of Eswatini (Hhohho, Lubombo, Manzini, and Shiselweni) was selected using convenience sampling.

### Sampling and recruitment

Participants were women meeting the following inclusion criteria: (i) attending one of the selected clinics, (ii) being aged between 25 and 59 years, and (iii) having no history of cervical cancer or hysterectomy. Consecutive sampling was used to select women until the target sample size was reached.

Women in the eligible age range were approached at the clinic, informed about the study and invited to participate. Those who agreed to participate were asked to sign a consent form (available in English and siSwati) before completing a paper copy of the study survey. A study log sheet was used to document the age of both consenting and non-consenting eligible women.

## Survey development and data collection

Three steps were implemented to produce survey items. The first step involved reviewing previous African research, World Health Organisation guidelines for screening and treating precancerous lesions for cervical cancer prevention [4], and Eswatini standardised cancer guidelines [5]. Twenty-five items were identified. The second step involved a review of identified items by six health behaviour experts. The panel of experts dropped one item about the benefits of cervical cancer screening because it was similar to another. An English version of the survey was developed based on these items.

The English survey was translated into siSwati using the forward-backward translation method [27]. A native bi-lingual siSwati translator independently translated the English survey into siSwati. This was followed by synthesising the forward translation, by the first author and the translator, into a satisfactory version. Afterwards, a second bilingual translator independently translated the synthesised forward translation into English. This translator was not permitted to see the original English version. After the backward translation was completed, both translators identified and resolved differences between the original and back-translated versions through discussions. A satisfactory version was reached after one round of backward-forward translation and discussions.

The 24-item survey was piloted before use in the current study. Bonett's formula [28] was used to calculate the minimum sample size required to test the reliability of the survey. Twenty-five eligible women were recruited to participate in the pilot study involving completing the siSwati version of the survey. To test the reliability, the internal consistency of the questionnaire was assessed by Cronbach's alpha coefficient. Content validity was evaluated by the panel of six health behaviour experts during survey development. Since the survey items were mainly derived from WHO and Eswatini screening guidelines, construct validity was not assessed.

As part of the feedback, women were asked to indicate whether: the instructions were easy to follow, the questions and/or response choices were easy to understand, whether they felt comfortable answering all the questions and whether they would be willing to complete a similar survey at future appointments (yes/no). Following piloting, the survey was adapted based on participants' feedback before use in this study. Based on the feedback of eight participants, minor language changes were made on two items.

A research assistant distributed self-administered pen-and-paper surveys to consenting participants. Each participant completed the survey privately, either while they waited to be seen by the nurse or at the end of their clinic visit. Survey completion took approximately 15 minutes.

## Measures

**Knowledge regarding cervical cancer screening.**   Based on literature from the African setting and in the developed world, four knowledge domains (risk factors of cervical cancer, benefits of screening, the meaning of screening results, and recommended screening intervals) were assessed. Knowledge regarding cervical cancer risk factors was evaluated using ten multiple-choice items [29–31]. Six multiple-choice questions were used to assess knowledge about

the Eswatini recommended screening intervals [4, 5]. Knowledge regarding the benefits of screening was measured using five items with true/false response options [10, 15, 16]. Three true/false items [15] were used to assess knowledge concerning the meaning of screening results. Each knowledge item was scored one if correct and zero if incorrect.

**Sociodemographic variables.** Four items were used to collect data on age, marital status, level of education, and availability of electricity in the participant's household [as an estimate of the participant's socioeconomic status [6]].

**Cervical cancer screening accessibility-related characteristics.** Two items (clinic visits in the past six months and travel time to clinic) [32, 33] were used to measure screening accessibility.

**Health status.** Two items assessed self-reported Human Immunodeficiency Virus (HIV) and Human Papilloma Virus statuses.

## Statistical analysis

We assessed selection bias by comparing the age distributions of eligible clinic attendees and study participants using Fisher's exact test. Cronbach's alpha coefficient was used to estimate the reliability of the knowledge items, and an alpha equal to or greater than 0.70 was considered satisfactory. Mean scores and ranges for overall knowledge and each knowledge domain were calculated for each participant. Individual item scores were added to calculate each participant's domain and overall knowledge scores. The minimum and maximum possible overall knowledge scores were 0 and 24, respectively. Participants' overall knowledge scores were described by their mean, standard deviation (SD) and range. The overall knowledge scores were used to construct an additional dichotomous knowledge variable (0 = scoring below the mean overall score and 1 = scoring equal to or above the mean overall score). Women scoring equal to or above the mean overall score were regarded as having relatively high cervical cancer screening-related knowledge [34]. The frequency and proportion of participants with relatively high cervical cancer screening-related knowledge was calculated.

Descriptive analyses were performed to assess participants' sociodemographic, cervical cancer screening accessibility-related characteristics, and health status-related characteristics. Mean, standard deviation (SD), and range were used to describe participants' age. Age was categorised into 25 – 35 years, 36 – 45 years, and 46 – 59 years. Frequencies and proportions were used to assess all other characteristics. The number and proportions of participants who correctly responded to each knowledge item (and 95% confidence intervals) were also calculated.

Linear regression was applied to examine associations between cervical cancer screening-related knowledge and participants' sociodemographic characteristics (sociodemographic, cervical cancer screening accessibility-related characteristics, and health status-related characteristics). Prior to conducting linear regression analysis, model assumptions were checked, including normality of residuals, homoscedasticity, multicollinearity and model fit. Linearity was not assessed since the model had no continuous independent variable. To check for normality of residuals, homoscedasticity visuals and statistical tests (the Shapiro-Wilk test and White's test, respectively) were conducted. The Pearson Chi-square tests were used to check the independence between categorical variables. In conjunction with residual analysis and model diagnostics, the R-squared statistic was used to assess the regression model fit.

Previous research and knowledge of clinical importance were used to determine the initial list of covariates to examine in regression analysis. As part of model building, univariate linear regression analysis was used to select variables to include in the first multivariable linear regression model. Covariates with a univariate $p$-value of $<0.25$ were considered for inclusion in the multiple regression model. A model with all selected covariates was fitted, after which

model reduction was assessed. Covariates that were no longer significant (at $p < 0.25$) in the multivariable model were tested for removal from the model. If the covariate's removal did not substantively change the remaining coefficients in the model by $> \sim 10\%$, the covariate was removed from the final model [35]. Likelihood ratio tests were used to decide between competing models. The significance level for multivariate analysis was set at a 0.05 threshold with 95% confidence intervals. Statistics and Data (STATA) software version 16 was used to conduct all statistical analyses.

## Sample size

The current study is a sub-study of a larger one reported elsewhere [36], which investigated non-adherence to cervical cancer screening recommendations among women in Eswatini. The aims of this larger study informed the sample size of 377 for the current study.

## Results

The Strengthening the Reporting of Observational Studies in Epidemiology (STROBE) [37] guided reporting in the current study. All clinics that were approached and invited consented. Out of 459 women approached, 416 met the eligibility criteria. Thirty-nine women declined to participate in the study due to either lack of interest in research or time to complete the survey. Therefore, our final sample of 377 participants gave us a consent rate of 91%. Response rates across clinics were as follows: clinic A – 96%, clinic B – 88%, clinic C – 91%, and clinic D – 89%. No differences between the age distributions of community health workers in Eswatini and study participants were found.

## Participants' characteristics

Participants' sociodemographic characteristics are shown in Table 1. A majority of the study participants: were 25 – 35 years old (234, 62%), were single (160, 42%), had secondary/ high school level of education (229, 61%), reported being HIV-negative (198, 53%), did not know whether they were HPV-negative or -positive (373, 99%), had electricity in their households (304, 81%), reported travel time to the clinic of 30 minutes or less (192, 51%), and had visited the clinic at least twice in the past six months (237, 63%).

## Cervical cancer screening-related knowledge

**Overall knowledge scores.** Cronbach's alpha coefficients for the different knowledge domains were estimated as follows; overall knowledge items (0.74), risk factors of cervical cancer items (0.77), benefits of screening items (0.78), the meaning of screening items (0.72), and recommended screening intervals items (0.65). The overall cervical cancer screening knowledge score ranged between eight and 21 (maximum possible = 24). The mean overall knowledge score was 16 (SD = 2.18). Two hundred and twenty-nine (61%) participants had relatively high knowledge scores (answered 80% or more questions correctly).

**Knowledge of risk factors.** The mean total knowledge score for cervical cancer risk factors among all participants was 7 (SD = 1.81), out of a possible range of 0 to 10. Birth control pill usage was the only risk factor correctly identified by less than half (143, 38%) of the study participants. Detailed results on knowledge of risk factors are shown in Table 2.

**Knowledge of benefits of screening.** Most study participants correctly answered most of the benefits of cervical cancer screening items (Table 2). The mean total knowledge score for benefits of screening among all participants was 4 (SD = 0.87), out of a possible range of 0 to 6.

**Table 1. Participants' characteristics (N=377).**

| Characteristics | n (%) |
|---|---|
| **Age** | |
| Mean (SD, range) | 35 (9.6, 25– 59) |
| 25 – 35 years | 234 (62) |
| 36 – 45 years | 76 (20) |
| 46 – 59 years | 67 (18) |
| **Marital status** | |
| Single | 160 (42) |
| Married | 151 (40) |
| Divorced/separated | 33 (9) |
| Widowed | 1 (0.3) |
| Living with a partner | 33 (9) |
| **Education** | |
| No formal education/primary school | 108 (29) |
| Secondary/high school | 229 (61) |
| Tertiary | 40 (11) |
| **Self-reported HIV-positive status** | 179 (47) |
| **Self-reported HPV status** | |
| HPV-positive | 4 (1) |
| HPV-negative | - |
| I don't know | 373 (99) |
| **Have electricity** | 304 (81) |
| **≤30 minutes travel time to clinic** | 192 (51) |
| **Clinic visits in the past six months** | |
| Never | 60 (16) |
| Once | 80 (21) |
| At least twice | 237 (63) |

Only 157 (42%) participants correctly identified the "cervical cancer screening can detect cancer, but only when the person has symptoms of cancer" item as false.

**Meaning of screening results.** The mean total knowledge score for benefits of screening among all participants was 3 (SD = 0.58), out of a possible range of 0 to 3. Of the three items, only one - "If screening shows abnormal changes in the cervix, this always means a woman has cervical cancer" - was correctly answered by 79% (n=299) of the participants.

**Knowledge of screening intervals.** The mean total knowledge score for the screening intervals domain among all participants was 2 (SD = 0.87), out of a possible range of 0 to 7. Few participants correctly identified Eswatini's recommended age at first (48, 13%) and last (115, 31%) cervical screening tests. Also, only a minority of the participants knew about differences in the frequency of screening for women according to HIV and HPV statuses. For example, only 5% (20) correctly answered items about the Eswatini recommended screening frequency for HIV-negative women and HPV-positive (or with unknown status).

## Associations between participants' characteristics and cervical cancer screening-related knowledge

None of the linear regression assumptions was violated. An R-squared value of 0.34 suggested that only 34% of the variance in the dependent variable is explained by the independent variables in our model. Age, marital status, HIV status, and travel time to the clinic met the criteria

**Table 2. Proportions of women correctly responding to items regarding four domains of cervical cancer and screening-related knowledge (N = 377).**

| Item assessing knowledge | Correct response | Number (%) giving a correct response | 95% CI |
|---|---|---|---|
| **Risk factors** | | | |
| Infection with human papillomavirus | True | 359 (95) | 93 – 97 |
| Having more than one sexual partner | True | 323 (87) | 82 – 89 |
| Being bewitched | False | 310 (82) | 78 – 86 |
| Having sex before the age of 16 | True | 307 (81) | 77 – 85 |
| Giving birth before the age of 16 | True | 305 (81) | 77 – 85 |
| Having reduced body immunity | True | 258 (68) | 63 – 73 |
| Living with Human Immunodeficiency Virus | True | 239 (63) | 58 – 68 |
| Smoking | True | 192 (52) | 46 – 56 |
| Giving birth more than once | True | 187 (50) | 44 – 55 |
| Use of birth control pills | True | 143 (38) | 33 – 43 |
| **Benefits of screening** | | | |
| The earlier cervical cancer is detected, the better the chance of recovery. | True | 364 (97) | 94 – 98 |
| Cervical screening reduces one's chance of dying from cervical cancer | True | 331 (88) | 84 – 91 |
| Screening can detect abnormal changes in the cervix before they become cancer. | True | 313 (83) | 79 – 87 |
| Screening can detect abnormal changes that have become cancer, even if the person has no symptoms. | True | 307 (81) | 77 – 85 |
| Cervical screening can detect cancer, but only when the person has cancer symptoms. | False | 157 (42) | 37 – 47 |
| **Meaning of screening results** | | | |
| If screening shows abnormal changes in the cervix, a woman may need more tests to determine whether she has cancer. | True | 357 (95) | 92 – 97 |
| If screening shows no abnormal changes in the cervix, a woman won't need to screen in the future. | False | 348 (92) | 89 – 95 |
| If screening shows abnormal changes in the cervix, this always means a woman has cervical cancer. | False | 299 (79) | 75 – 83 |
| **Cervical screening interval** | | | |
| How often should a woman have cervical cancer screening if they are: HIV positive? | Annually | 341 (90) | 87 – 93 |
| After the cervical screening, when does a woman get their results: | Immediately | 277 (73) | 69 – 78 |
| At what age should women have their last cervical screening test? | 59 years | 115 (31) | 26 – 35 |
| At what age should women have their first cervical screening test? | 25 years | 48 (13) | 10 – 17 |
| How often should a woman have cervical cancer screening if they are: HIV negative and HPV-negative? | Every 3 years | 28 (7) | 5 – 11 |
| How often should women have cervical cancer screening if they are: HIV negative and HPV-positive or status is unknown? | Every 2 years | 20 (5) | 3 – 7 |

($p \leq 0.25$) for inclusion in the adjusted linear regression model. HIV status and travel time to the clinic were the only variables significantly associated with cervical cancer screening-related knowledge. Compared to HIV-positive participants, HIV-negative participants had 0.61 times lower cervical cancer screening knowledge scores (β = -0.39, 95% CI: -0.56, -0.19, $p = 0.03$), assuming all other variables in the model were held constant. Participants who travelled more than 30 minutes to the clinic had 0.3 times lower cervical cancer screening knowledge scores (β = -0.70, 95% CI: -1.15, -0.25, $p < 0.01$) compared to participants who travelled less than 30 minutes to the clinic (Table 3).

## Discussion

The current study presents information on cervical cancer screening knowledge and associated factors among a sample of women 25 – 59 years old in Eswatini. This study examined other pertinent knowledge domains besides previously reported knowledge about cervical cancer risk factors. These included recommended screening intervals and the benefits and meaning

**Table 3. Associations between cervical cancer screening-related knowledge and participants' sociodemographic characteristics (N=377).**

| Woman characteristic | Overall knowledge score | | | |
|---|---|---|---|---|
| | Unadjusted β coefficient (95% CI) | p-value | Adjusted β coefficient (95% CI) | p-value |
| **Age** | | | | |
| 25 – 35 years | R | | R | |
| 36 – 45 years | 0.65 (0.09,1.22) | 0.02* | 0.55 (-0.06,1.16) | 0.08 |
| 46 – 59 years | 0.10 (-0.49,0.69) | 0.73 | 0.13 (-0.56,0.81) | 0.72 |
| **Marital status** | | | | |
| Single | R | | R | |
| Married | 0.39 (-0.09,0.88) | 0.11* | 0.37 (-0.15,0.90) | 0.16 |
| Divorced/separated/widowed | -0.10 (-0.92,0.72) | 0.81 | -0.38 (-1.32,0.55) | 0.42 |
| Living with a partner | -0.34 (-1.16,0.48) | 0.41 | -0.44 (-1.25,0.36) | 0.28 |
| **Education** | | | | |
| No formal education/primary school | R | | | |
| Secondary/high school | -0.17 (-0.67,0.33) | 0.50 | - | - |
| Tertiary | -0.23 (-1.02,0.57) | 0.57 | - | - |
| **HIV status** | | | | |
| HIV-positive | R | | R | |
| HIV-negative | -0.39 (-0.83,-0.22) | 0.03* | -0.39 (-0.56,-0.19) | 0.03** |
| **Have electricity** | | | | |
| Yes | R | | | |
| No | -0.22 (-0.78,0.33) | 0.43 | - | - |
| **Travel time to the clinic** | | | | |
| ≤30 minutes | R | | R | |
| >30 minutes | -0.53 (-0.97,-0.09) | 0.02* | -0.70 (-1.15,-0.25) | <0.01** |
| **Clinic visits in the past six months** | | | | |
| Never | R | | | |
| Once | 0.25 (-0.48,0.98) | 0.50 | - | - |
| At least twice | 0.39 (-0.35,0.89) | 0.39 | - | - |

\* - statistically significant *p-value* at α = 0.25

\** - statistically significant *p-value* α = 0.05

of cervical cancer screening results. Results indicate that most women had higher screening knowledge levels. Cervical cancer screening knowledge scores of HIV-negative participants were significantly lower than those of HIV-positive participants. Also, the knowledge levels of women who travelled more than 30 minutes to the clinic were lower than those who travelled less than 30 minutes.

## Cervical cancer screening-related knowledge

Almost two-thirds (61%) of the participants in the current study had relatively high overall knowledge scores. However, several studies in sub-Saharan Africa suggest that women generally lack information about cervical cancer screening [38–40]. Accordingly, a systematic review of sub-Saharan studies reported low knowledge and awareness about cervical cancer and screening as a critical barrier to participation in cervical cancer screening [41].

The relatively high overall knowledge in the current study may be related to the sample's somewhat higher education levels; 71% had at least secondary or high school level education. Studies in the Democratic Republic of Congo [42] and Ethiopia [43, 44] concluded that educational attainment was positively associated with women's knowledge about cervical cancer

screening. In addition, the relatively high cervical cancer screening knowledge could also be attributed to sampling women attending primary healthcare clinics. These women may be expected to have more opportunities to access cervical cancer information than those in the community [32].

While the overall knowledge of cervical cancer risk factors was relatively high, knowledge about specific risk factors was limited. For example, only 38% of the study participants correctly associated using the birth control pill and the risk of developing cervical cancer. While women may need to be educated about this risk factor, health education messages addressing this risk should be carefully crafted to avoid undermining the use of this birth control method which has helped reduce other problems such as unplanned pregnancies, abortions and maternal morbidity and mortality [45]. More importantly, health education messages should address the importance of consistent and correct use of condoms in preventing pregnancy and male-to-female genital HPV transmission [46].

Most participants (58%) believed screening could detect cancer only when a woman had cancer symptoms. Previous African studies have reported similar findings [47, 48]. This misconception may mean that screening is perceived as a diagnostic test rather than a screening test and could result in delays in screening [49]. Therefore, it may be beneficial to incorporate differences between screening and diagnostic tests in future health education messages.

One of the positive findings of this study was that most participants had higher scores of knowledge regarding the meaning of screening results. This may mean that participants are exposed to high-quality information from health providers regarding the meaning of cervical cancer screening [50]. Personal experience with screening may also increase women's knowledge about screening [51].

Women's knowledge of Eswatini's recommended screening intervals, particularly the age for starting and frequency of screening according to HIV and HPV statuses, was poor. Two previous studies reporting similar results suggested that this knowledge gap was likely related to poor dissemination and communication of screening interval information by primary healthcare clinic nurses [52, 53]. In line with this, some previous African studies have reported that healthcare workers generally have insufficient knowledge about cervical cancer screening age and interval among healthcare workers [10, 54]. Therefore, it would be necessary for educational programs (to increase knowledge about recommended screening intervals) to target both health providers and women.

## Associations between participants' characteristics and cervical cancer screening-related knowledge

The current study found statistically significant associations between HIV status, travel time to the healthcare clinic and knowledge about cervical cancer screening. HIV-negative participants had significantly lower knowledge levels than HIV-positive participants. Similar results were reported in studies conducted in Zambia [55], the Lao People's Democratic Republic [56], and the USA [57]. One reason for this finding could be that, compared to HIV-negative women, HIV-positive women may have better access to information about cervical cancer and screening [55]. Since 2018, the US Government, through PEPFAR, has supported the Government of Eswatini in cervical cancer screening and treating precancerous lesions for HIV-positive women. HIV-positive women are prioritised through this program with health education and cervical cancer screening [58]. Also, HIV-positive women are more likely to visit health facilities (where most health education occurs) as they seek life-long HIV treatment and care [55]. In the process, their chances of being informed about cervical cancer and screening may increase.

Participants who travelled more than 30 minutes to the clinic had lower cervical cancer screening knowledge scores than those who travelled less than 30 minutes to the clinic. This likely reflects reduced access to health care (including health education) for individuals who travel for extended periods to reach the nearest healthcare clinic [59]. While no previous study has reported similar findings, this finding should nonetheless be considered by health programs in Eswatini when tailoring education interventions to increase community cervical cancer screening-related knowledge. Community-based health education implemented by community health workers is an example of an intervention that may increase cervical cancer screening-related knowledge in the community [60].

## Strengths and limitations

The study had a high response rate (estimated at 91%), increasing the chances of the results being representative of our target sample. Also, the current study examined crucial cervical cancer screening knowledge domains (benefits of screening, meaning of screening results, and screening intervals) that previous Eswatini studies have overlooked. Several limitations of the present study should be acknowledged. Firstly, our results may not be generalisable to the general population of Eswatini women, as the participants were only recruited from primary healthcare clinics. Therefore, results need to be interpreted with caution. Second, women's HIV statuses were self-reported. Women may have underreported being HIV positive to present themselves in a socially desirable way. Third, due to participant burden concerns, it was impossible to assess all possible correlates of knowledge. Fourth, the current study focused on cervical cancer screening knowledge. While knowledge has been shown to be correlated with screening uptake in past research (7, 8, 62), interventions focussing on knowledge alone are ineffective in improving screening [61]. This suggests that knowledge may be a necessary but not sufficient condition for cervical cancer screening participation [62, 63]. The need for future well-controlled intervention studies to examine the impact of interventions focussed on knowledge and addressing other barriers to screening participation is indicated. Finally, while our model identified statistically significant relationships, only 34% of the variance in the dependent variable was explained. This remaining 66% may be due to random error or other factors not captured by our model. Despite meeting model assumptions, these limitations suggest avenues for future research. Increasing the sample size and exploring potential independent variables could strengthen the reliability and generalizability of our findings. Additionally, incorporating qualitative methods could further clarify the complex dynamics at play.

## Conclusion

While overall knowledge about cervical cancer screening was relatively high among the study participants, there were gaps in knowledge regarding specific risk factors of cervical cancer and when to begin and how often to screen. These findings may inform future educational programs to create and sustain an accurate understanding of cervical cancer screening in Eswatini communities. Educational programs must advocate for and target HIV-negative women and women who travel for extended periods to reach the nearest healthcare clinic. Future studies involving larger samples from the community may help confirm and explain why travel time matters in cervical cancer screening knowledge levels.

## Acknowledgments

The authors would like to thank Ms Siphosethu Mamba for her assistance with collecting the study data. The authors are also grateful to Dr Christopher Oldmeadow for the statistical advice.

## Author Contributions

**Conceptualization:** Phinda G. Khumalo, Mariko Carey, Lisa Mackenzie, Rob Sanson-Fisher.

**Formal analysis:** Phinda G. Khumalo.

**Methodology:** Phinda G. Khumalo, Mariko Carey, Lisa Mackenzie, Rob Sanson-Fisher.

**Project administration:** Phinda G. Khumalo.

**Supervision:** Mariko Carey, Lisa Mackenzie, Rob Sanson-Fisher.

**Visualization:** Rob Sanson-Fisher.

**Writing – original draft:** Phinda G. Khumalo.

**Writing – review & editing:** Mariko Carey, Lisa Mackenzie, Rob Sanson-Fisher.

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
