## [Decision Letter · Decision Letter 0]

16 Nov 2022

PONE-D-22-25805A Cross-sectional Study Reporting the Level of Knowledge Regarding Cervical Screening among Women in EswatiniPLOS ONE

Dear Dr. Khumalo,

Thank you for submitting your manuscript to PLOS ONE. After careful consideration, we feel that it has merit but does not fully meet PLOS ONE’s publication criteria as it currently stands. Therefore, we invite you to submit a revised version of the manuscript that addresses the points raised during the review process.

Please submit your revised manuscript by Dec 31 2022 11:59PM. If you will need more time than this to complete your revisions, please reply to this message or contact the journal office at plosone@plos.org. Please include the following items when submitting your revised manuscript:A rebuttal letter that responds to each point raised by the academic editor and reviewer(s). You should upload this letter as a separate file labeled 'Response to Reviewers'.A marked-up copy of your manuscript that highlights changes made to the original version. You should upload this as a separate file labeled 'Revised Manuscript with Track Changes'.An unmarked version of your revised paper without tracked changes. You should upload this as a separate file labeled 'Manuscript'.

We look forward to receiving your revised manuscript.

Kind regards,

Gulzhanat Aimagambetova

Academic Editor

Journal Requirements

We will update your Data Availability statement to reflect the information you provide in your cover letter."

Reviewers' comments:

Reviewer's Responses to Questions

**Comments to the Author**

1. Is the manuscript technically sound, and do the data support the conclusions?

Reviewer #1: No

Reviewer #2: Yes

2. Has the statistical analysis been performed appropriately and rigorously? 

Reviewer #1: No

Reviewer #2: Yes

3. Have the authors made all data underlying the findings in their manuscript fully available?

Reviewer #1: No

Reviewer #2: Yes

4. Is the manuscript presented in an intelligible fashion and written in standard English?

Reviewer #1: Yes

Reviewer #2: Yes

5. Review Comments to the Author

Reviewer #1: The study aimed to assess the knowledge of cervical screening in Eswatini women. The study results suggest overall respondents had a high knowledge level, and some factors were independently associated with the knowledge. I have several comments and concerns that need clarification or additional considerations. Please see them below:

Check the sentence in the lines 85-87

The authors could be more explicit in discussing how this study differs from previous research.

There two aims but there are not reflected in the manuscript title and abstract.

How do ineligible healthcare clinics differ from the included clinics?

What were exclusion criteria?

Were all women recruited to take part in the study? What were response rates across healthcare clinics? What type of sampling technique was utilized?

The survey development section needs more clarification. How many items initially were selected? (the authors may provide the list of identified items in the supplementary materials section) How did experts agree on the final 29-item survey (please describe the process)? Were the survey in English or siSwati? If yes, who and how the survey was translated? Could the authors provide more information on how the survey was modified after the pretesting (they may include before and after survey versions)? What type of pretesting technique was utilized? What was the rationale for selecting only 25 participants? Have the authors checked internal reliability and validity of the survey when pretested it?

What theoretical framework was applied to answer the research question? The theoretical framework may help with explaining the rationale for selecting some covariates for the analysis.

It is not clear how the overall knowledge score was calculated. What was the rationale for dichotomizing the overall knowledge score (why this cut-off were selected)?

The authors mentioned that the Cronbach’s alpha was calculated to assess the internal reliability. I wonder whether inter-item correlation and potentially explanatory factor analysis was performed?

In the analysis part, the authors stated that “The potential for introducing confounding bias was also considered in addition to significance. No confounding effects were found.” Could they elaborate more what do mean by this statement and how it was performed?

Also, it is not clear what was the rationale for using 0.25 threshold for selection of variables for the multivariable analysis? If a variable passed the threshold in the univariate analysis, what happened next? All variables were included in the model? What about clinically or epidemiologically important variables?

Have the authors checked the linear regression assumptions? Multicollinearity? Model fit?

Prior conducting the study, have the authors calculated the required sample size for this study? If not, maybe it would be interesting to determine the statistical power of the analysis?

In Table 3, have p-value estimates for multi-level categorical variables were corrected for multiple pair-wise comparison tests?

I am concerned about the robustness of the study results given some of the important independent factors for the knowledge are missing or not included in the survey (e.g., socio-economic status, previous history of gynecological diseases, history of cervical cancer screening). I am also concerned about the novelty of the research, because the knowledge may not actually correlated with the uptake. If data permits, I would be interested to see if the hypothesis can be tested and results presented to make a stronger conclusion.

I believe the limitations section also needs some improvements based on the comments raised in this review.

Reviewer #2: Dear Authors,

Thank you for this opportunity to read and review your manuscript. In covers an important topic as Cervical cancer remains a huge medical and public health issue, especially in low- and middle-income countries. Cervical cancer prevention programs are well developed now, however, in some parts of the world it is still not properly implemented. The manuscript is clearly written in professional, unambiguous language. The reference list is up-to-date. However, some corrections should be made before acceptance and publishing.

1. The manuscript title should be rephrased: current version “A Cross-sectional Study Reporting the Level of Knowledge Regarding Cervical Screening among Women in Eswatini”. Suggested - The Level of Knowledge Regarding Cervical Cancer Screening among Women in Eswatini: A Cross-sectional Study. Or any other that will sound more clear.

2. The abstract represents the study very well. However, it should be revised. The academic writing style of the abstract and the manuscript in general should be improved. It is better if the text is rewritten to be from a third parity. Example: current version ‘We conducted a cross-sectional study involving three hundred and seventy-seven women aged 25 to 59 selected from four primary healthcare clinics in Eswatini.” Suggested - A cross-sectional study involving three hundred and seventy seven women aged 25 to 59 selected from four primary healthcare clinics in Eswatini was conducted.

3. The introduction part gives a clear understanding of the study rationale. However, please rethink whether you need subheadings in the introduction part.

4. A short notes about the country, Eswatini, would be very useful for a potential reader to understand specifics o the country. One paragraph noting the country location, population, income (World Bank) – low- or middle income, specifics of the healthcare system pointing on the cervical cancer screening and vaccination (if it is covered by the government or not, which is partially already mentioned). This information will definitely help potential readers to get into the problem discussed in the manuscript.

I would recommend removing the subheadings form the introduction, it will improve the text comprehension.

5. Aims should not be separated from the introduction part, but presented as a last paragraph in the introduction. It will sound better without numbering (1 and 2), but as a plane text. Please always refer to the PLOS ONE Guide for authors.

6. The methods part includes all required data. However, as in the introduction part, there are too many subsections, some of them could be merged. Example: subsection “Sample” (line 120) should be merged with subsection “Sampling and recruitment” (line 124). Please revise the methods section accordingly.

7. Please mention in the methods part if the STROBE guideline for cross-sectional studies was used. If not, then why?

8. What criteria were used to select clinics for inclusion in the study?

9. When choosing a clinic to be included in the study, was the location of the clinic considered?

10. Has the WHO questionnaire been adapted considering local peculiarities?

11. The results section presents a very interesting and valuable data, supported by clear tables and description.

12. In general, the discussion part is interesting, however, requires improvement to meet the style appropriate for research manuscripts. The first paragraph of the discussion should provide to a potential reader the study rationale in brief. Please find below the suggestions how the discussion part should be restructured:

Discussion

1.1 Rationale of the study (why it was done)

1.1.1 Main findings of the study

1.1.2 What makes our study unique

1.1.3 What it adds to what we already know

1.2 Subject of the discussion

Comparison of our results with neighboring countries, with countries of the same

development levels (income), with developed high-income countries). Agreement and

disagreement with the studies compared. Suggested similar articles to compare with (from countries of the similar income) - doi: 10.1200/GO.20.00619; doi: 10.1177/17455065211004135; doi: 10.1371/journal.pone.0261203

1.4 Sum up of the study, study strengths and limitations

13. Please ensure the manuscript is written according to the PLOS One guidelines https://journals.plos.org/plosone/s/submission-guidelines

6. PLOS authors have the option to publish the peer review history of their article (what does this mean?). If published, this will include your full peer review and any attached files.

Reviewer #1: No

Reviewer #2: No

---

## [Author Response · Author response to Decision Letter 0]

4 Aug 2023

Dear Editorial Office,

Re: “PONE-D-22-25805: A Cross-sectional Study Reporting the Level of Knowledge Regarding Cervical Screening among Women in Eswatini”

Thank you for allowing us to submit a revised draft of our manuscript. We are grateful for the insightful comments on our manuscript, which have improved it. We have incorporated most of the suggestions made by the reviewers. Those changes are highlighted in yellow in the text. Please see below for a point-by-point response to the reviewers’ comments and concerns. 

REVIEWER COMMENTS

Reviewer #1: 

Check the sentence in lines 85-87. The authors could be more explicit in discussing how this study differs from previous research.

In lines 79 – 87, we have modified the description of previous research and highlighted how this research provides important methodological improvements.

There are two aims, but they are not reflected in the manuscript title and abstract.

Changes have been made to the title to reflect the second aim. Please see page 1.

How do ineligible healthcare clinics differ from the included clinics? What were the exclusion criteria?

Clinics providing care to fewer than 50 women aged 25-59 per week and those attending only to HIV-positive women were ineligible to participate. This information has been added on page 5, lines 121 to 122. 

Were all women recruited to take part in the study? 

A consecutive sample of women who met the inclusion criteria was asked to participate in the study. This information has been on page 6, lines 129 – 130.

What were response rates across healthcare clinics?

Response rates across clinics were as follows; clinic A – 96%, clinic B – 88%, clinic C – 91%, and clinic D – 89%. This information has been on page 10, lines 242 – 244.

What type of sampling technique was utilized?

Consecutive sampling was used to select women until the target sample size was reached. Please see page 6, lines 129 – 130.

The survey development section needs more clarification. How many items initially were selected? (the authors may provide the list of identified items in the supplementary materials section)

The survey development section has been modified on pages 6 to 7, lines 137 to 166. An English version of the survey is also attached to this response letter.

How did experts agree on the final 29-item survey (please describe the process)? 

The process is described on page 6, lines 137 to 143.

Were the survey in English or siSwati? If yes, who and how the survey was translated?

The process is described on pages 6 to 7, lines 144 to 152.

Could the authors provide more information on how the survey was modified after the pretesting (they may include before and after survey versions)?

Information is provided on page 7, lines 161 to 166.

What type of pretesting technique was utilized? 

The technique is described on page 7, lines 154 to 170.

What was the rationale for selecting only 25 participants? 

Bonett's formula was used to calculate the minimum sample size required to test for Cronbach's Alpha- coefficient (please see page 7, lines 153 – 154). In addition, previous research recommends sample sizes between 24 and 50 (Sim and Lewis, 2012; Julious, 2005).

Sim J, Lewis M. The size of a pilot study for a clinical trial should be calculated in relation to considerations of precision and efficiency. J Clin Epidemiol 2012; 65:301-308.

Julious SA. Sample size of 12 per group rule of thumb for a pilot study. Pharm Stat 2005; 4:287-291.

Have the authors checked internal reliability and validity of the survey when they pretested it?

As stipulated on page 7, lines 153 to 160, Cronbach’s alpha coefficient was used to estimate the reliability of the knowledge items. Content validity was assessed by six health behaviour experts during survey development. Given that the survey was mainly derived from WHO and Eswatini screening guidelines, construct validity was not assessed.

What theoretical framework was applied to answer the research question? The theoretical framework may help with explaining the rationale for selecting some covariates for the analysis.

Based on previous research, we expected age, marriage, education, HPV-positive status, socioeconomic status (based on electricity availability), and the number of clinic visits in the past six months to correlate positively with knowledge, while the distance to the nearest clinic would be negatively correlated. This information has been included in the revised manuscript on pages 4 to 5, lines 99 to 103.

It is not clear how the overall knowledge score was calculated. 

The description of how we calculated the overall score has been modified on page 8, lines 194 to 197.

What was the rationale for dichotomizing the overall knowledge score (why this cut-off were selected)?

Dichotomising the knowledge score allowed labelling/describing participants as having or not having an outcome (higher level of knowledge in this case) to facilitate a better understanding of our sample. This was only done for descriptive analysis purposes. In regression analysis, we used the knowledge score as a continuous variable to avoid losing information, decreasing statistical power and increasing the probability of type I error.

The authors mentioned that the Cronbach’s alpha was calculated to assess the internal reliability. I wonder whether inter-item correlation and potentially explanatory factor analysis was performed?

The survey items were taken from the WHO guidelines, which are deemed credible. Also, a panel of experts deemed the survey items relevant. For these two reasons, we did not perform inter-item correlation. Performing explanatory factor analysis was not part of the study’s aims. Also, given our sample size, this analysis may have been insufficiently powered.

In the analysis part, the authors stated that “The potential for introducing confounding bias was also considered in addition to significance. No confounding effects were found.” Could they elaborate more what do mean by this statement and how it was performed?

We have modified our statement on adjusting confounding in the analysis phase on page 10, lines 228 to 233.

Also, it is not clear what was the rationale for using 0.25 threshold for selection of variables for the multivariable analysis?

As a rule of thumb, a liberal threshold is used in univariate analysis when selecting variables for the multivariable analysis. According to Hosmer and Lemeshow, and Sperandei, using traditional levels of significance, such as 0.05, can fail to identify variables associated with the independent variable.

Hosmer DW, Lemeshow SJAlr. Interpretation of the fitted logistic regression model. 2000;2:47-90.

Sperandei S. Understanding logistic regression analysis. Biochem Med (Zagreb). 2014;24(1):12-8.

If a variable passed the threshold in the univariate analysis, what happened next?

Please see page 10, lines 224 to 230, for details on model building.

All variables were included in the model? 

Please see a description of variables included in the multivariable regression model on page 15, lines 286 to 287.

What about clinically or epidemiologically important variables?

As described on page 9, lines 221 to 222 an initial list of covariates to be examined in regression analysis was determined using previous research and knowledge of clinical importance. Likelihood ratio tests were used to decide between competing models.

Have the authors checked the linear regression assumptions? Multicollinearity? Model fit?

A statement summarising the statistical analyses conducted to check the linear regression assumptions can be found on page 9, lines 214 to 220. A summary of results from these analyses is presented on page 15, lines 285 to 286.

Prior conducting the study, have the authors calculated the required sample size for this study? If not, maybe it would be interesting to determine the statistical power of the analysis?

Sample size determination is described on page 10, lines 238 to 240.

In Table 3, have p-value estimates for multi-level categorical variables were corrected for multiple pair-wise comparison tests?

Given our sample size (377), we believe there was no indication for correcting for multiple pair-wise comparisons. A rule of thumb by Green (1991) in Tabachnick and Fidell (2014) suggests that to avoid the problem of multiple testing, the sample size should be greater or equal to 104 + m, where m is the number of predictors. Given 20 predictors in the current study, 104 + m equals 124. 

Tabachnick, B. G., Fidell, L. S., & Ullman, J. B. (2007). Using multivariate statistics (Vol. 5, pp. 481-498). Boston, MA: pearson.

I am concerned about the robustness of the study results given some of the important independent factors for the knowledge are missing or not included in the survey (e.g., socio-economic status, previous history of gynecological diseases , history of cervical cancer screening)

We selected correlates that had strong backing from previous research to avoid the risk of multiple testing associated with including many correlates. We may have mixed studies suggesting other variables. As a result, we have included the potential to omit potential confounder/important covariates as one of the study’s limitations on page 20, lines 384 to 392.

I am also concerned about the novelty of the research, because the knowledge may not actually correlated with the uptake. If data permits, I would be interested to see if the hypothesis can be tested and results presented to make a stronger conclusion.

In the limitations section (page 20, lines 386 to 392), it is now made clear that knowledge is a necessary but not a sufficient condition for screening participation. The need for future well-controlled intervention studies to determine what might help increase screening participation is indicated.

I believe the limitations section also needs some improvements based on the comments raised in this review.

The limitations section has been revised.

Reviewer #2: 

The manuscript title should be rephrased: current version “A Cross-sectional Study Reporting the Level of Knowledge Regarding Cervical Screening among Women in Eswatini”. Suggested - The Level of Knowledge Regarding Cervical Cancer Screening among Women in Eswatini: A Cross-sectional Study. Or any other that will sound more clear.

The manuscript title has been modified. Please see page 1.

The abstract represents the study very well. However, it should be revised. The academic writing style of the abstract and the manuscript in general, should be improved. It is better if the text is rewritten to be from a third party. Example: current version ‘We conducted a cross-sectional study involving three hundred and seventy-seven women aged 25 to 59 selected from four primary healthcare clinics in Eswatini.” Suggested - A cross-sectional study involving three hundred and seventy seven women aged 25 to 59 selected from four primary healthcare clinics in Eswatini was conducted.

The abstract and manuscript have been rewritten in the third person.

The introduction part gives a clear understanding of the study rationale. However, please rethink whether you need subheadings in the introduction part. I would recommend removing the subheadings from the introduction, it will improve the text comprehension.

Subheadings have been removed.

A short notes about the country, Eswatini, would be very useful for a potential reader to understand specifics of the country. One paragraph noting the country location, population, income (World Bank) – low- or middle income, specifics of the healthcare system pointing on the cervical cancer screening and vaccination (if it is covered by the government or not, which is partially already mentioned). This information will definitely help potential readers to get into the problem discussed in the manuscript.

Information about Eswatini has been added on page 5, lines 107 to 120.

Aims should not be separated from the introduction part, but presented as a last paragraph in the introduction. It will sound better without numbering (1 and 2), but as a plane text. Please always refer to the PLOS ONE Guide for authors.

As suggested, aims are now presented in the last paragraph of the introduction section. Please see pages 4 to 5, lines 93 to 103.

The methods part includes all required data. However, as in the introduction part, there are too many subsections, some of them could be merged. Example: subsection “Sample” (line 120) should be merged with subsection “Sampling and recruitment” (line 124). Please revise the methods section accordingly.

As suggested methods section has been modified.

Please mention in the methods part if the STROBE guideline for cross-sectional studies was used. If not, then why?

A statement on the STROBE guideline has now been included on page 10, lines 238 to 239. The STROBE checklist is also attached to this response.

What criteria were used to select clinics for inclusion in the study?

Information on inclusion criteria is presented on page 5, lines 121 to 122.

When choosing a clinic to be included in the study, was the location of the clinic considered?

As indicated on page 5, lines 121 to 123, the location (region) of the clinic was considered when we selected clinics. Since all primary healthcare clinics are in rural areas, rurality was not considered.

Has the WHO questionnaire been adapted considering local peculiarities?

No WHO questionnaire was used. Instead, WHO guideline recommendations were expressed in question form and used to assess knowledge among women. WHO guideline recommendations account for local peculiarities, especially socioeconomic status, such that there are specific recommendations for poor resource countries and developed countries. Countries adopt (from the WHO recommendations) guidelines that would work for their local situations. 

In general, the discussion part is interesting, however, requires improvement to meet the style appropriate for research manuscripts. The first paragraph of the discussion should provide to a potential reader the study rationale in brief. Please find below the suggestions how the discussion part should be restructured:

As suggested, the discussion section has been modified. Please see pages 17 to 20.

We look forward to hearing from you regarding our submission and responding to any further questions and comments you may have.

Kind regards,

Phinda Goodwill Khumalo

---

## [Decision Letter · Decision Letter 1]

13 Dec 2023

PONE-D-22-25805R1Cervical Cancer Screening Knowledge and Associated Factors among Eswatini Women: A Cross-sectional StudyPLOS ONE

Dear Dr. Khumalo,

Thank you for submitting your manuscript to PLOS ONE. After careful consideration, we feel that it has merit but does not fully meet PLOS ONE’s publication criteria as it currently stands. Therefore, we invite you to submit a revised version of the manuscript that addresses the points raised during the review process.

We look forward to receiving your revised manuscript.

Kind regards,

Gulzhanat Aimagambetova

Academic Editor

PLOS ONE

Journal Requirements:

Reviewers' comments:

Reviewer's Responses to Questions

**Comments to the Author**

1. If the authors have adequately addressed your comments raised in a previous round of review and you feel that this manuscript is now acceptable for publication, you may indicate that here to bypass the “Comments to the Author” section, enter your conflict of interest statement in the “Confidential to Editor” section, and submit your "Accept" recommendation.

Reviewer #1: All comments have been addressed

2. Is the manuscript technically sound, and do the data support the conclusions?

Reviewer #1: Yes

3. Has the statistical analysis been performed appropriately and rigorously? 

Reviewer #1: No

4. Have the authors made all data underlying the findings in their manuscript fully available?

Reviewer #1: No

5. Is the manuscript presented in an intelligible fashion and written in standard English?

Reviewer #1: Yes

6. Review Comments to the Author

Reviewer #1: All my previous comments were addressed. I have some minor comments that authors may consider addressing to strengthen their manuscript:

In the lines 286-287, it says that the “…The goodness of fit of the regression 287 model was estimated at 34%...”. I wonder what type of test was used? How to interpret this result?

In the sample size calculation section, it seems the formula for prevalence sample size calculation was used. I wonder what parameters were used to calculate the required sample size? What were the assumptions?

7. PLOS authors have the option to publish the peer review history of their article (what does this mean?). If published, this will include your full peer review and any attached files.

Reviewer #1: No

---

## [Author Response · Author response to Decision Letter 1]

19 Feb 2024

JOURNAL REQUIREMENTS

- To the best of our knowledge, none of the cited papers in our manuscript have been retracted. However, due to difficulty in identifying the publication journal of "Knowledge of cervical cancer and patronage of cervical cancer screening services among female health workers in Kumasi, Ghana 2011" by Adageba et al. (reference 54), we opted to remove this reference in line 347. The information it provided is adequately covered by the remaining references within the same topic area.

- Three references in the reference list have been reformatted to comply with the journal's specific formatting requirements. These changes are highlighted on page 24.

- An additional reference has been added to the reference list and cited on line 234.

REVIEWER COMMENTS

Reviewer #1: 

In the lines 286-287, it says that the “…The goodness of fit of the regression model was estimated at 34%...”. I wonder what type of test was used? How to interpret this result?

- In lines 285-287, we have modified this statement and mentioned that “An R-squared value of 0.34 suggested that only 34% of the variance in the dependent variable is explained by the independent variables in our model.” 

- We acknowledge that given our results, about 66% of the variance is unexplained by the model and could be due to random error or other factors not included in the analysis. We have since included this as one of the limitations of our study in lines 393-399. 

In the sample size calculation section, it seems the formula for prevalence sample size calculation was used. I wonder what parameters were used to calculate the required sample size? What were the assumptions?

- The current study is a sub-study of a larger one (Reference 37) investigating non-adherence to cervical cancer screening recommendations among women in Eswatini. To estimate the proportion of non-adherent women in the target population, the larger study aimed for a sample size of 385 using a 95% confidence interval and a 5% margin of error for an expected proportion of 50% (formula: n = [p̂ × (1 - p̂) × z2)]/MOE2. While the current study aimed to recruit 385 women, only 377 consented to participate.

---

## [Decision Letter · Decision Letter 2]

5 Mar 2024

Cervical Cancer Screening Knowledge and Associated Factors among Eswatini Women: A Cross-sectional Study

PONE-D-22-25805R2

Dear Dr. Phinda Khumalo,

We’re pleased to inform you that your manuscript has been judged scientifically suitable for publication and will be formally accepted for publication once it meets all outstanding technical requirements.

Kind regards,

Gulzhanat Aimagambetova

Academic Editor

PLOS ONE

Additional Editor Comments (optional):

Reviewers' comments:

Reviewer's Responses to Questions

**Comments to the Author**

1. If the authors have adequately addressed your comments raised in a previous round of review and you feel that this manuscript is now acceptable for publication, you may indicate that here to bypass the “Comments to the Author” section, enter your conflict of interest statement in the “Confidential to Editor” section, and submit your "Accept" recommendation.

Reviewer #1: All comments have been addressed

2. Is the manuscript technically sound, and do the data support the conclusions?

Reviewer #1: Yes

3. Has the statistical analysis been performed appropriately and rigorously? 

Reviewer #1: Yes

4. Have the authors made all data underlying the findings in their manuscript fully available?

Reviewer #1: No

5. Is the manuscript presented in an intelligible fashion and written in standard English?

Reviewer #1: Yes

6. Review Comments to the Author

Reviewer #1: (No Response)

7. PLOS authors have the option to publish the peer review history of their article (what does this mean?). If published, this will include your full peer review and any attached files.

Reviewer #1: No

---

## [Editor Report · Acceptance letter]

27 Mar 2024

PONE-D-22-25805R2 

PLOS ONE

Dear Dr. Khumalo, 

I'm pleased to inform you that your manuscript has been deemed suitable for publication in PLOS ONE. Congratulations! Your manuscript is now being handed over to our production team.

Kind regards, 

on behalf of

Dr. Gulzhanat Aimagambetova 

Academic Editor

PLOS ONE